# Immobilized *Stenotrophomonas maltophilia* KB2 in Naproxen Degradation

**DOI:** 10.3390/molecules27185795

**Published:** 2022-09-07

**Authors:** Danuta Wojcieszyńska, Judyta Klamka, Ariel Marchlewicz, Izabela Potocka, Joanna Żur-Pińska, Urszula Guzik

**Affiliations:** Institute of Biology, Biotechnology and Environmental Protection, Faculty of Natural Science, University of Silesia in Katowice, Jagiellońska 28, 40-032 Katowice, Poland

**Keywords:** biodegradation, immobilization, naproxen, *Stenotrophomonas*

## Abstract

Immobilization is a commonly used method in response to the need to increase the resistance of microorganisms to the toxic effects of xenobiotics. In this study, a plant sponge from *Luffa cylindrica* was used as a carrier for the immobilization of the *Stenotrophomonas maltophilia* KB2 strain since such a carrier meets the criteria for high-quality carriers, i.e., low price and biodegradability. The optimal immobilization conditions were established as a temperature of 30 °C, pH 7.2, incubation time of 72 h, and an optical density of the culture of 1.4. The strain immobilized in such conditions was used for the biodegradation of naproxen, and an average rate of degradation of 3.8 µg/hour was obtained under cometabolic conditions with glucose. The obtained results indicate that a microbiological preparation based on immobilized cells on a luffa sponge can be used in bioremediation processes where it is necessary to remove the introduced carrier.

## 1. Introduction

Naproxen, a bicyclic propionic acid derivative, is a non-steroidal anti-inflammatory drug (NSAID). Due to its treatment properties and low toxicity, it is often chosen in pain treatment. Its action mechanism is based on the inhibition of both cyclooxygenase isoforms COX 1 and COX 2—the enzymes responsible for synthesizing prostaglandins, prostacyclin, and thromboxane from arachidonic acid [1]. Naproxen is absorbed quickly from the gastrointestinal tract, has a long duration of action, and is also able to diffuse into the synovial fluid. Hence, it is often recommended for patients with osteoarthritis, especially those burdened with cardiovascular diseases [2]. The availability of over-the-counter naproxen and its numerous advantages translate into its strong position in the pharmaceutical market. Every year, thousands of tons of naproxen are produced by pharmaceutical companies and then purchased and used by consumers [3]. In the human body, naproxen undergoes a slight transformation to desmethylnaproxen (30%) and partly to glucuronide derivatives (65%), and in these forms is excreted in the urine [4]. Naproxen and its metabolites thus get into treatment plants. Depending on the region, the observed naproxen concentrations in the influents to wastewater treatment plants vary significantly. Guedes-Alonso et al. [5] showed that, in the influent of the treatment plant in Gran Canaria, Spain, the concentration of naproxen was 521.7 µg/L. On the other hand, in Algeria, the observed concentrations were significantly lower and ranged from 1.22 to 9.59 µg/L [6]. In turn, significantly higher, reaching up to 1370 µg/L, was observed in southwest India’s Manipal sewage treatment plant [7]. However, the microbiome of wastewater treatment plants often does not completely degrade the drug. Hence, they have been observed in effluents [6,7,8]. For example, in a wastewater treatment plant in Algeria, the effluents have naproxen concentrations ranging from undetectable to 0.334 µg/L [6]. On the other hand, the sewage treatment plant’s effluent in Manipal showed naproxen in concentrations ranging from 0.02 to 70.1 µg/L [7]. This way, they reach surface waters and pose a threat to aquatic organisms. Currently, naproxen is detected in rivers and lakes on all continents in the concentration range of ng/L to μg/L. Hence, it is crucial to find tools capable of degrading naproxen with high efficiency [2]. Moreover, non-steroidal anti-inflammatory drugs, including naproxen, usually go to sewage treatment plants in the company of other xenobiotics, such as phenols, derivatives of aromatic acids, or metal compounds. Therefore, it is crucial that the strains that support the degradation of NSAIDs can degrade a broader range of xenobiotics or at least be resistant to their toxic effects. The *Stenotrophomonas maltophilia* KB2 strain (GenBank DQ230920.1) belongs to the microorganisms capable of degrading phenolic derivatives, such as nitrophenols and chlorophenols, as well as selected organic acids. Moreover, it is characterized by the ability to synthesize dioxygenases of various types, cleaving the aromatic ring, and catalysis in the presence of heavy metal salts [9,10]. Moreover, Wojcieszyńska et al. [11] observed that this strain could degrade naproxen in the presence of an additional carbon source. However, the performance of this degradation has fluctuated in the range of 40–78%. It is well-known that immobilized microorganisms show a higher efficiency of pollutant degradation in bioremediation processes due to resistance to increased concentrations of toxic compounds. It was found that immobilization increased the stability of plasmid DNA containing, among other things, genes encoding enzymes involved in the degradation of xenobiotics [12,13].

To support wastewater treatment plants, immobilized microorganisms are increasingly used. The use of immobilized cells has been shown to have numerous advantages over traditional free cell systems. Immobilized microorganisms can be used continuously or repeatedly in batch processes without the risk of washing them out of the bioreactor. Immobilization ensures the stability of biochemical transformations and allows for their intensification by increasing the bioavailability of the substrate and specific enzymatic and respiratory activity, which reduces the costs of processes [12,14,15]. Many methods have been developed to immobilize cells using natural and synthetic carriers [12,14,15,16]. Li et al. [17] used the fungus *Phanerochaete chrysosporium* immobilized on sawdust in the continuous degradation of naproxen in a countercurrent seepage bioreactor. During the continuous operation of the bioreactor for 165 days, the loss of naproxen to an undetectable level was obtained. The research was carried out in non-sterile conditions, modelling the operation of sewage treatment plants. Despite this, no bacterial infection of the system and no reduction in fungal activity were observed, which the authors attribute to the protective effect of sawdust. More and more works are carried out using natural materials, especially waste from the agricultural industry. Although they show lower mechanical strength than synthetic ones, they are more environmentally friendly, less toxic to microorganisms, and provide a higher rate of diffusion of reagents [12,18]. One of the most promising organic carriers of natural origin is the plant sponge *Luffa cylindrica*. In 1993, it was used for the first time to immobilize the cells of the red microalga *Porphyridium cruentum* [19]. The plant sponge is a lignocellulosic biopolymer consisting of cellulose (60%), hemicelluloses (30%), and lignins (10%), with an open, fibrous structure, high porosity (79–93%), a pore volume of 21–29 cm^3^/g, low density (0.02–0.04 g/cm^3^), and high specific surface (850 m^2^/m^3^). It has been shown that acid functional groups, including carboxyl, enol, lactone, and phenol groups, are present in large amounts on the fibers’ surface [19]. Such a structure allows for the adsorption of large quantities of microorganisms and their growth in the pore space and reduces diffusion resistance due to the possibility of quick and direct contact of cells with the medium. In addition, this polymer is stable over a wide range of pH and temperatures. It can be used in long-lasting and repeatable processes, is easy to use, and is relatively cheap, making it an exciting carrier material [3,20]. Dzionek et al. [14] used this carrier to immobilize the naproxen-degrading strain *Bacillus thuringiensis* B1(2015b). The preparation prepared in this way was introduced into a lab-scale trickling filter, previously inoculated with biomass obtained from the Imhoff tank from the sewage treatment plant in Krupski Młyn–Ziętek in Poland. During the operation of the trickling filter, the synergistic influence of the autochthonous microflora on the naproxen biodegradation that was performed by the immobilized B1(2015b) cells was observed. It resulted in 1 mg/L naproxen degradation within 15 days [14].

Although immobilized microorganisms are more resistant to xenobiotics’ toxic effects and changing environmental conditions, immobilization does not continually improve degradation efficiency. Therefore, the research aimed to determine the optimal conditions for the immobilization of the *Stenotrophomonas maltophilia* KB2 strain on the luffa plant sponge and answer whether the immobilization would improve the naproxen degradation capacity of the KB2 strain.

## 2. Results and Discussion

The ability of microorganisms to colonize solid surfaces is an important feature in determining effective immobilization. Adsorption is a simple, fast, cheap, and non-toxic method for microorganisms. Due to the diversity of microhabitats within the biofilm, the developed network of communication and exchange of genetic material between cells, and the properties of the extracellular polymeric substance (EPS) secreted by them, microorganisms are characterized by an increased tolerance to unfavorable environmental conditions, including the stress of starvation, temperature changes, and environmental reactions and the presence of high concentrations of xenobiotics [12,13,21]. Immobilized microorganisms appear to be particularly useful in bioremediation systems due to their increased survival, stability of the created environment, and high catalytic activity compared to microorganisms introduced by bioaugmentation [12,14]. However, the development of a biocatalyst ensuring the high efficiency of this process depends on the strong binding of microbial cells to the matrix and the quality of the biofilm formed by them. Hence, there is a need to optimize the immobilization conditions for a given strain and carrier material [14,22].

### 2.1. Optimization of the KB2 Strain Immobilization on the Luffa Sponge

#### 2.1.1. Influence of Temperature

The *Stenotrophomonas maltophilia* KB2 is capable of biofilm formation and shows high self-aggregation capacity. It was also found that the strain is capable of twitching, swarming, and swimming, indicating a locomotive apparatus—flagella and type IV pili [23,24,25]. These structures are involved in the adhesion of bacteria to solid surfaces [26]. However, the KB2 strain cannot synthesize exopolysaccharides. Hence, the formation of a stable biofilm probably depends on the appropriate interactions between flagella or pili of the adsorbed cells [27,28].

Research on the influence of temperature on the immobilization process showed that this parameter has no significant influence on bacteria immobilization. When used in the immobilization of a culture with an OD_600_ 1.2, the observed number of immobilized cells of the KB2 strain did not differ significantly at different temperatures, despite the observed differences in the amount of biomass on the support (Figure 1A).

It is probably that the stress of hunger plays a significant role in the adhesion of the strain on the support. Numerous data in the literature show that, without an external carbon source, microorganisms react by stopping cell division and growth and switching to maintenance [29,30]. Then, the dimensions of cells decrease due to an increase in the consumption of endogenous growth substrates, which leads to their dwarfing, or reduction division, which increases the number of cells with a simultaneous decrease in their biomass [31,32]. Consequently, this leads to an increase in the favorable surface-to-volume ratio and an increase in cell aggregation [33,34], which colonize solid surfaces more efficiently than single cells and can be retained in the pore spaces of the carrier [35,36]. The high efficiency of immobilization on a porous sponge matrix is probably the result of the ability of the KB2 strain to self-aggregate. In addition, it is postulated that changes occurring in bacterial cells during starvation lead to the acquisition of cross-resistance by microorganisms to other stress factors, such as a change in culture temperature, the high osmotic pressure of the solution, oxidative stress, exposure to ultraviolet radiation, or the presence of acids and solvents (acetone, ethanol) [37,38]. It was found that the regulation of the stress response to hunger and related cross-resistance is carried out at the level of gene expression with the participation of the alternative sigma factor σ^S^, encoded by the *rpoS* gene [32], which is a positive regulator of the expression of genes involved in the formation of biofilm [39]. The *rpoS* gene has been found in numerous species of Gram-negative bacteria from the Gammaproteobacteria class, including *Escherichia coli, Pseudomonas aeruginosa, Pseudomonas putida, Erwinia carotovora*, and *Salmonella* sp. [40,41]. The presence of σ^S^ homologues was found in *Xanthomonas campestris* [42]. Hence, due to their belonging to the same order, their occurrence is also probable in the genus *Stenotrophomonas*. The controlled expression of σS cross-resistance to low and high temperatures probably allows the KB2 strain to bind to the plant sponge matrix with statistically similar efficiency, expressed in the number of live cells immobilized on the support, although the optimal temperature for environmental *Stenotrophomonas maltophilia* strains is in the range of 20–30 °C [43].

The enhancement of the σ^S^ effect on the condition of the KB2 cells subjected to starvation stress at the suboptimal temperatures of the culture may explain the observed differences between the amount of biomass on the support. A statistically significant decrease in the value of this parameter was observed at low temperatures (4 and 10 °C) compared to the biomass content at optimal temperatures (20–30 °C), for which an estimated similar number of immobilized cells amounted to 10^7^−10^8^ CFU/mL. (Figure 1A). On the other hand, the increase in the number of immobilized cells at higher temperatures (35 and 40 °C) concerning the number of cells introduced into the culture, which was not accompanied by a significant increase in the weight of the biofilm on the support compared to the increase in biomass at the other tested temperatures (Figure 1A), proves the possibility of reduction divisions in the bacterial population. The adaptive effect of temperature on the morphology of starved cells was noted by Santander and Biosca [44] in the Gram-negative bacterium *Erwinia amylovora*. It should be emphasized that the measurement of the biofilm content on the carrier should only be taken into account in the evaluation of the holistic efficiency of the immobilization process because the mature biofilm, apart from living cells, also includes dead and dormant cells [21].

The quality of the formed biofilm was determined by measuring the catalytic activity of the immobilized cells. The total enzymatic activity (TEA) results obtained for the immobilized cells of strain KB2 at different culture temperatures did not significantly differ (Figure 1B). It was observed that the increase in temperature above 25 °C promotes the increased metabolic activity of the biofilm. The observed high values of deviations of metabolic activity measurements for the biofilm formed at 4 and 10 °C and the lack of correlation between the higher number of cells immobilized at 35 and 40 °C and their enzymatic activity show that the KB2 strain is sensitive to temperatures outside the optimum (20–30 °C). It is likely, however, that maintaining a certain minimum endogenous metabolism caused by starvation stress of immobilized KB2 cells, regardless of the culture temperature, is possible due to the protective role of the carrier [29,45]. Considering the above, 30 °C was selected as the optimal temperature for the immobilization of the KB2 strain.

#### 2.1.2. Influence of Incubation Time

Studies on the effect of incubation time on the immobilization of *Stenotrophomonas maltophilia* KB2 strain have shown that it can colonize a plant sponge within 24 h of cultivation with a yield of 91%. Analyzing the number of immobilized cells and the biofilm content on the carrier in the consecutive days of culture (Figure 2A), it was observed that after 48 h of incubation, the values of these parameters were significantly lower, by 27 and 24%, respectively, compared to the results obtained after 24 h.

This allowed the conclusion that some cells detached from the carrier between 24 and 48 h of culture. This phenomenon is probably due to the shear forces acting on the aeration of the culture by agitation. These forces play an essential role in the initial adhesion of microorganisms to solid surfaces by influencing the conformational structure of adhesins and the motor properties of adsorbing cells [46,47,48]. They also regulate the dispersion processes of mature biofilm [49]. The results obtained during the immobilization of the KB2 strain indicate that cells at the stage of reversible adhesion could be detached from the carrier due to the presence and motor functions of flagella/pili. It is also possible that cells in the biofilm interact the least with the matrix surface, which may depend on the number and the strength of interaction between the pili of individual cells within the microcolony and may additionally be caused by the lack of a protective extracellular matrix [24,50]. The metabolic condition of immobilized cells, which was significantly low after 24 h of immobilization of the KB2 strain under fasting conditions, has an important influence on forming a stable biofilm (Figure 2A) [12,13].

The formation of biofilm by the KB2 strain on a plant sponge was a dynamic process, and the stability of the immobilization system was achieved after 72 h of cultivation. The number of immobilized cells after 72 h of incubation was in the range of 7.0 × 10^7^−1.4 × 10^8^ CFU/mL. These differences indicate that, apart from the colonization of the carrier by new cells, a heterogeneous structure could have developed within the existing biofilm. The cells of the deeper layers assumed the VBNC (viable but non-culturable) phenotype or died out. Cells with the VBNC phenotype are unable to form colonies on bacteriological media, although they show metabolic activity. It is a defense strategy characteristic of bacteria subjected to long-term stress of hunger that is found, among others, in *Escherichia coli, Helicobacter pylori*, some strains of the genus *Vibrio*, and *Pseudomonas* [51]. It has been shown that, in biofilms formed under oligotrophic conditions, VBNC cells appear in the layers closest to the carrier surface due to the formation of a nutrient gradient [21]. The lack of dry weight loss of the biofilm with fewer colonies grown on agar plates suggests that, when the KB2 strain is immobilized after 72 h of incubation in the absence of an external carbon source, the appearance of VBNC cells is highly probable.

An increase in the total enzymatic activity of the biofilm formed on the luffa sponge during 72 h of incubation was observed (Figure 2B). It is in opposition to the studies by Dzionek et al. [22]. They noticed that the immobilization of *Bacillus thuringiensis* B1 (2015b) on a polyurethane sponge in a medium devoid of a carbon source was associated with a gradual decrease in the TEA value. This was caused by the systematic use of spare materials and the reduction in metabolic activity of immobilized cells to a minimum level. The studied strain is probably susceptible to the lack of an external carbon source. During the first 24 h of hunger, it uses most of the stored energy substrates needed, e.g., for adaptation to changed osmotic and pH conditions, as well as for active movement and formation of a biofilm on the support [29]. It is also possible that the cells of the KB2 strain were in the stationary growth phase before introduction into the mineral medium, in which the *rpoS* gene was activated, and the phenotype developed identically to that of cells subjected to hunger stress [32]. Biofilm formation is a defense strategy as it allows the cells of the strain to achieve higher metabolic activity under conditions of a long-lasting fasting period [13,52]. In summary, the obtained results indicate that 72 h is the optimal time to generate a stable and metabolically active biofilm of the KB2 strain on a plant sponge matrix.

#### 2.1.3. Effect of Initial Culture Density

The KB2 strain’s ability to colonize the plant sponge increases with the increase in the culture optical density. The effectiveness of immobilization in relation to the amount of inoculum introduced into the culture was 80% (OD 0.2); 86% (OD 0.4, 0.6, 0.8); 87% (OD 1.0); 88–91% (OD 1.2); and 108–113% (OD 1.4). The increase in the number of immobilized bacteria showed a very strong positive correlation with the increase in the biomass content on the support using denser and denser cultures (Figure 3A).

The observed relationships confirm that the introduction of an increasing inoculum of bacteria into the culture increases the probability of achieving a true quorum, i.e., the minimum required number of cells, which under given conditions of local distribution and diffusion of signaling molecules will cause a QS response [53].

A significant increase in the enzymatic activity of cells immobilized on a plant sponge was demonstrated using OD 1.0–1.4 (Figure 3B), which results from the colonization of the carrier with a greater number of bacteria. It may also be due to exposure to higher concentrations of the diffusible signal factor (DSF) in the bacterial population, which is part of the QS.

#### 2.1.4. Impact of pH

Changes in the pH of the culture medium significantly impact the initial adhesion of cells, as they are associated with a change in the surface charge of the solids contained in it, which is of particular importance for the interaction between hydrophilic surfaces. It has been shown that both the plant sponge and bacteria of the species *Stenotrophomonas maltophilia* are characterized by a high degree of surface hydration [54,55,56]. It was also found that the zero point of charge (pHPZC) of the plant sponge is 5.2 [57], and the dominant functional groups are the acid groups: carboxyl, enol, lactone, and phenol [19]. The pHPZC of *Stenotrophomonas maltophilia* bacteria has a value of 11. The possible cause of the positive charges on the surface is the lack of charge on lipopolysaccharide particles and the presence of small amounts of proteins in the outer membrane of cells [54].

The pH of the environment significantly influenced the colonization of the carrier by the cells of the KB2 strain. The effectiveness of the immobilization process in terms of the number of cells initially introduced into the culture (OD600 1.2) was the lowest at pH 9 (59%), followed by pH 8 and 5 (71 and 78%, respectively), and was highest at 7.2 (86%) (Figure 4A).

Due to the presence of positive surface charges in the entire range of the tested pH, the KB2 strain adsorbed to the plant sponge effectively at a pH where the surface of the sponge has a negative charge, as evidenced by relatively high values of biomass on the carrier and the number of immobilized bacteria at pH above 5 (Figure 4A).

In the case of the highest pH values (8; 9), the observed decrease in the number of cells immobilized on the carrier may be associated with a gradual reduction in the dominance of positive charges on the surface of bacterial cells as it approaches the pH value corresponding to their zero-charge point. It consequently reduces the attractiveness of binding with an increasingly negatively charged carrier matrix [58]. The decrease in the number of immobilized cells and the biomass content on the support was observed at the zero-charge point of the plant sponge. This confirms the importance of electrostatic interactions in the interaction between the hydrophilic cell surfaces of the KB2 strain and the luffa sponge [57].

It is likely that other types of interactions, e.g., hydrophobic interactions, also take part in the initial adhesion of the KB2 strain. It allows bacterial cells to colonize the plant sponge at a pH where electrostatic interactions are unfavorable due to the increase in the dominance of positive charges on its surface (pH 3–4) and the absence of a charge (pH 5) [59]. Starvation and acidification stress may increase the hydrophobicity of bacterial cells [34,60]. Despite the high degree of hydration of *Stenotrophomonas maltophilia* bacteria [54], they can form bonds with hydrophobic groups of sponge fibers, mainly contained in the structure of the lignin, which is a component of the sponge [19,57]. In addition, the motor structures of microorganism cells [61,62], also found in the KB2 strain, also play an important role in overcoming electrostatic repulsion in the initial adhesion [24,25]. In fibrous carrier materials, anomalies in the number of immobilized cells may be caused by the retention of their aggregates in the carrier pore spaces [63].

The effect of pH on the endogenous metabolism of immobilized cells, expressed as total enzymatic activity, is hardly noticeable. A slight increase in metabolic activity against a low number of cells immobilized on the carrier was observed at pH 8 and 9 (Figure 4B), which indicates the need to regulate intracellular pH and prevent cytoplasm alkalization by mechanisms such as K^+^(Na^+^)/H^+^ antiporters activity or membrane transporters of drugs with the function of substrate/H^+^ antiporters that increase the activity of ATP synthases to import protons into the cell with simultaneous ATP synthesis or increase the metabolic production of acid molecules through the process of amino acid deamination [64,65,66]. In summary, pH 7.2 was determined as the optimal pH of the culture medium for the immobilization of the KB2 strain due to the high values of dry weight and the number of live, immobilized cells. This results from the correct proportion between the number of positive charges on the surface of bacterial cells and the negative charges of the sponge plant at a given pH value.

### 2.2. Naproxen Degradation by Immobilized KB2 Strain

#### 2.2.1. Naproxen Biodegradation

To date, few strains of bacteria and fungi capable of degrading naproxen have been identified [11,67,68]. The most frequently observed intermediates are desmethylnaproxen and hydroxynaproxen, which only show the drug’s transformation, not its complete decomposition [2,69]. One of the few described strains for which the degradation route by hydroxylation and ring cleavage to 3,4-dicarboxybenzene acetic acid has been described is the KB2 strain [2,11]. Due to the high resistance of naproxen to biological degradation, the observed efficiency of degradation by pure bacterial strains is unsatisfactory [11,70,71]. Moreover, the resulting intermediates are often more toxic than the parent substance, limiting the bacteria’s growth [2]. A study by Dzionek et al. [15] indicates that immobilization may be a simple solution to improve degradation efficiency. In addition, it allows the use of immobilized strains in subsequent degradation cycles. The luffa sponge used in the presented work is a cheap and straightforward carrier. The KB2 strain immobilized on it can be used in bioremediation processes to remove difficult-to-degrade compounds, which the KB2 strain exhibits. In the presented work, the ability of immobilized cells of the KB2 strain to decompose one of the most difficult-to-degrade NSAIDs—naproxen—was tested. A high dose of 6 mg/L was used in the studies. No naproxen degradation was observed in monosubstrate conditions (Figure 5), but such conditions do not occur in the environment.

On the other hand, under cometabolic conditions with glucose as the carbon source, only 1.3 mg/L remained after 14 days (Figure 5). Naproxen was degraded at an average rate of 13.99 µg/h. The CFU/µL value on the fifth day of cultivation, obtained after the disintegration of the carrier, was 4062.5 CFU/µL. Although the KB2 strain did not show any better degradation abilities after immobilization than free cells [11], its immobilization enables the use of such a preparation many times. At high concentrations of naproxen, which is common in the environment along with accompanying pollutants, the carrier constitutes a protective barrier for bacteria [12].

#### 2.2.2. SEM Visualization of Immobilized Bacteria on a Luffa Sponge during Naproxen Degradation

The plant sponge with attached cells of the KB2 strain, which were immobilized under optimal conditions, was observed by SEM. Microphotographic analysis showed that the immobilization of the strain was successful (Figure 6).

The microscopic image shows irregular and multilayered structures forming a biofilm on the sponge fibers. During the cultivation period, the overgrowth of the free spaces of the sponge was observed. Similar observations were also made in the SEM analysis of the immobilized strains: *Bacillus thuringiensis* B1 (2015b), *Planococcus* sp. S5, and *Pseudomonas moorei* KB4 [3,14,72].

## 3. Materials and Methods

### 3.1. Immobilization of KB2 Strain

The Stenotrophomonas maltophilia KB2 strain was grown in nutrient broth for 24 h at 30 °C, 130 rpm. Then, the culture was centrifuged (5000 rpm, 4 °C, 20 min), and the resulting suspension was used for immobilization. Immobilization of the KB2 strain was carried out on a natural Luffa cylindrica sponge (Pixel Perfect, Poznań, Poland). Pieces weighing 100 mg were cut from the outer walls of the sponge seed and dried to a constant weight. A carrier weighing 500 mg was introduced into 100 mL of mineral salt medium (MSM) and sterilized at 121 °C, 1.2 atmospheres for 20 min [22,73].

The effect of temperature on the immobilization process was investigated by incubating the carrier in the medium with the KB2 strain at the temperatures of 4 °C, 10 °C, 20 °C, 25 °C, 30 °C, 35 °C, and 40 °C for 72 h at 130 rpm.

The influence of the incubation time on the immobilization of the KB2 strain was determined by incubating the carrier in the MSM with the strain for 24, 48, and 72 h at a temperature of 30 °C and 130 rpm.

The evaluation of the influence of the optical density of the culture on the adsorption capacity of the KB2 strain was investigated by incubating the carrier in MSM with a bacterial inoculum with an OD600 of 0.2; 0.4; 0.6; 0.8; 1.0; 1.2, and 1.4, for 72 h at 30 °C and 130 rpm.

The analysis of the influence of the pH of the medium on the effectiveness of the immobilization of the KB2 strain was carried out by adjusting the pH of the MSM before inoculating with bacterial cells to values of 5, 6, 7, 7.2, 7.6, 8, and 9. In the medium prepared this way, the carrier was incubated with the tested strain for 72 h at 30 °C and 130 rpm.

### 3.2. Biochemical Analysis

#### 3.2.1. Assay of Enzymatic Activity of Immobilized Bacteria

The metabolic activity of immobilized microorganisms was measured using the fluorescein diacetate method (FDA). For this purpose, a single plant sponge with immobilized cells of the KB2 strain was placed in 8 mL of phosphate buffer at pH 7.6, and 100 μL of fluorescein diacetate at a concentration of 4.8 mmol/L was added to the carrier. The sample was incubated for 60 min in the dark at 30 °C, 130 rpm. The reaction was stopped with 2 mL of acetone, and the level of fluorescein absorbance was measured. Concentrations were determined based on a calibration curve obtained using standard fluorescein solutions. The enzymatic activity of bacteria was calculated as fluorescein concentration per gram of dry weight of the carrier per 1 h of incubation [22,74].

#### 3.2.2. Determination of the Dry Mass of Bacteria

The dry weight of the immobilized bacteria was determined by comparing the dry weight of the carrier before and after the immobilization process, after drying at 105 °C for 2 h. Based on the obtained results, the percentage share of biofilm in the total mass of the carrier was calculated using Formula (1) [22,75].
% biofilm = (dry mass of the carrier after immobilization/dry mass of the carrier before immobilization) × 100%(1)

#### 3.2.3. Determination of the Number of Colony-Forming Units

To determine the number of immobilized bacterial cells on the luffa sponge, a single piece of sponge with bacteria was placed in 25 mL of physiological saline and homogenized at 16,000 rpm for 20 s with a homogenizer and sonicated for 20 s at 25 kHz in an ultrasonic bath. The resulting suspension of released bacteria was diluted in a series of saline dilutions and plated on nutrient agar plates. After 48 h of incubation at 30 °C, the number of colony-forming units (CFU) was calculated, and the number of cells immobilized on the matrix was determined according to Formula (2) [76].
n = ñ_j_a^−j^a_p_^−1^(2)
where n—the actual number of colony-forming units; ñ_j_—mean number of colony-forming units in J-th dilution; a^−j^—dilution factor; a_p_^−1^—sample volume applied to the agar plate.

### 3.3. Biofilm Analysis by Scanning Electron Microscopy

To confirm the presence of the attached cells on the carrier surface during naproxen degradation, scanning electron microscopy (SEM) was used. The carrier with immobilized bacteria was prepared for imaging with SEM using 3% *v*/*v* glutaraldehyde incubation (fixative, 24 h) and subsequent ethanol dehydration (30, 50, 70, 80, 90, 95, and 100% *v*/*v*, each for 15 min). The samples were subsequently critical-point-dried in the Leica EM CPD300 Automated Critical Point Dryer (Leica Microsystems, Vienna, Austria), mounted on aluminum stubs with double-sided adhesive carbon-tape and sputter-coated in a Pelco SC-6 Sputter Coater (Ted Pella Inc., Redding, CA, USA) with a thin film of gold to improve the electrical conductivity of the sample surface. After processing, samples were imaged using the Hitachi SU8010 field emission scanning electron microscope (FESEM) (Hitachi High-Technologies Corporation, Tokyo, Japan) at 10 kV accelerating voltage with a secondary electron detector (SED) and at a working distance (WD) of 3–300 μm.

### 3.4. Naproxen Degradation Study

Degradation of naproxen in monosubstrate and cometabolic systems was performed in 500 mL Erlenmeyer flasks containing 250 mL of the MSM [11] inoculated with 5 pieces of the luffa sponge colonized by bacteria. The control was 250 mL of sterile MSM with sterile carriers. Naproxen was added to each flask to obtain a final concentration of 6 mg/L, and all cultures were incubated with shaking at 30 °C for 15 days. For studies on the cometabolic degradation of naproxen, 0.5 mg/L glucose was added. If the complete degradation of the suitable growth substrate was observed, a successive dose of glucose was introduced. All cultures were grown in triplicates.

To study the degradation of naproxen, 2 mL samples were taken periodically from the culture medium and centrifuged (6000× *g*, 15 min). The concentration of naproxen in the culture supernatant was determined according to Wojcieszyńska et al. [11].

### 3.5. Statistical Analysis

Statistical analysis was performed using the STATISTICA 13 PL software (TIBCO Software Inc., Palo Alto, CA, USA) based on the one-way ANOVA test, with the adopted significance level of *p* < 0.05. The least significant difference test of NIR was performed to show differences between the conditions of the immobilization process (*p* < 0.05).

## 4. Conclusions

Studies have shown that starvation conditions favor the adsorption of the KB2 strain on a plant sponge, but they significantly affect the condition of immobilized microorganisms, which is reflected in the total enzymatic activity. The temperature does not significantly influence the effectiveness of the KB2 strain immobilization process. This is likely due to changes in microbial cells during fasting and the development of cross-resistance to low and high temperatures. The biofilm formation by the KB2 strain is a dynamic process conditioned by the action of shear forces during the aeration of the culture by shaking. A more extended incubation period leads to the formation of a stable and metabolically active biofilm on the support. In the process of initial adhesion of the KB2 strain to the plant sponge, apart from electrostatic interactions, other types of interactions, e.g., hydrophobic interactions, probably take part. A significant decrease in the adsorption properties of the strain was noted at the zero-charge point of the plant sponge (pH 5) and pH close to the zero-charge point of bacterial cells (pH 8 and 9). The metabolic activity of the immobilized cells was dependent on the pH of the culture medium. Degradation studies with the immobilized KB2 strain indicated naproxen degradation under cometabolic conditions. This indicates the application potential of the constructed biopreparation, but it requires further research in semi-industrial conditions.

## Figures and Tables

**Figure 1 molecules-27-05795-f001:**
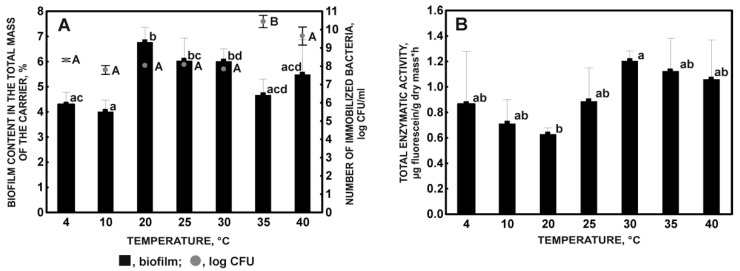
Biofilm content in the total mass of the sponge and the number of immobilized bacteria (**A**), and the total enzymatic activity of the immobilized microorganisms (**B**) in relation to the temperature after 72 incubation hours. Different letters indicate statistically significant differences (ANOVA, *p* < 0.05).

**Figure 2 molecules-27-05795-f002:**
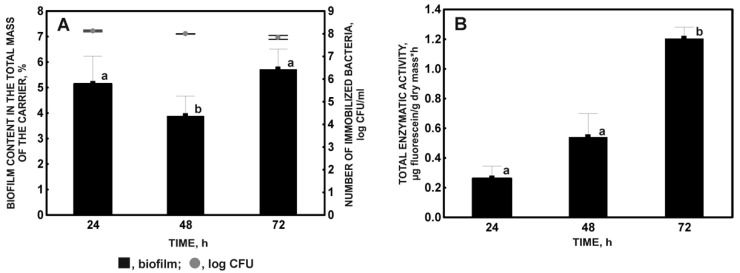
Biofilm content in the total mass of the sponge and the number of immobilized bacteria (**A**), and the total enzymatic activity of the immobilized microorganisms (**B**) in relation to the incubation time. Different letters indicate statistically significant differences (ANOVA, *p* < 0.05).

**Figure 3 molecules-27-05795-f003:**
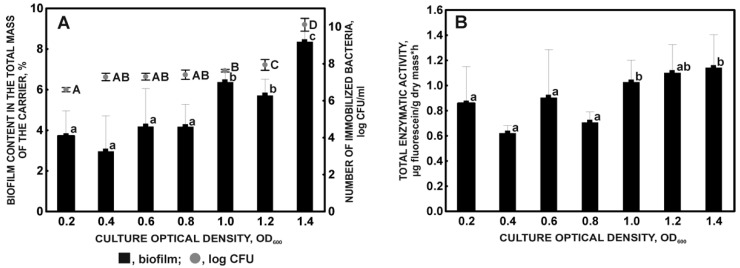
Biofilm content in the total mass of the sponge and the number of immobilized bacteria (**A**), and the total enzymatic activity of the immobilized microorganisms (**B**) in relation to the culture optical density after 72 incubation hours. Different letters indicate the existence of statistically significant differences (ANOVA, *p* < 0.05).

**Figure 4 molecules-27-05795-f004:**
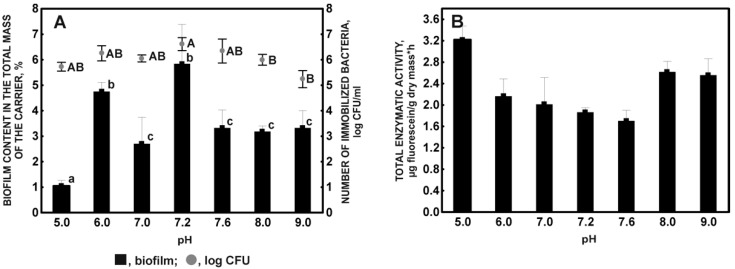
Biofilm content in the total mass of the sponge and the number of immobilized bacteria (**A**), and the total enzymatic activity of the immobilized microorganisms (**B**) in relation to the pH after 72 incubation hours. Different letters indicate the existence of statistically significant differences (ANOVA, *p* < 0.05). Results without any lowercase letters are statistically equal at a significance level *p* < 0.05.

**Figure 5 molecules-27-05795-f005:**
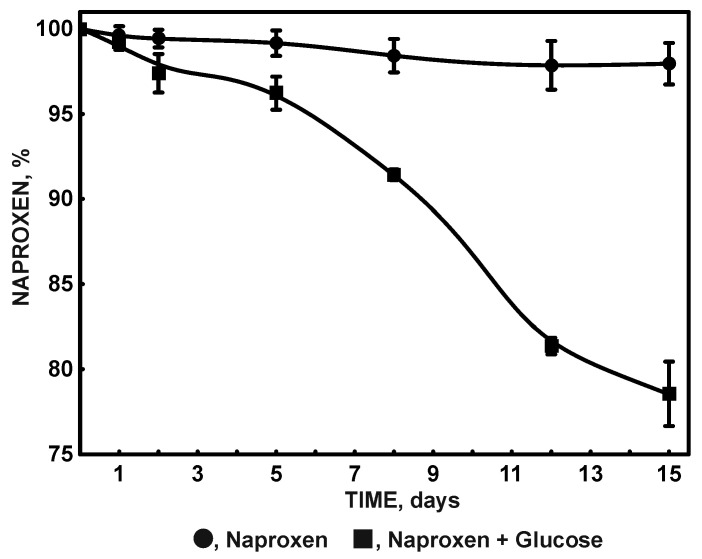
Degradation of 6 mg/L naproxen by strain KB2 without additional carbon source and with 0.5 mg/L glucose as a simple carbon source.

**Figure 6 molecules-27-05795-f006:**
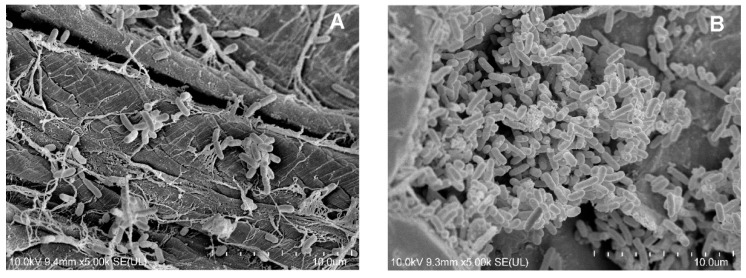
SEM microphotographs of *Stenotrophomonas maltophilia* KB2 cells immobilized in optimal conditions at 4 (**A**) and 15 (**B**) cultivation days. The working distance was 9.4 mm and 9.3 mm for the (**A**) and the (**B**) microphotographs, respectively.

## Data Availability

The data presented in this study are available on request from the corresponding author. The data are not publicly available due to privacy restrictions.

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
