# Peer review of "Immobilized Stenotrophomonas maltophilia KB2 in Naproxen Degradation"

_molecules, 2022, doi:10.3390/molecules27185795_

Round 1

Reviewer 1 Report

The MS entitled "Immobilized Stenotrophomonas maltophilia KB2 in naproxen degradation" by Wojcieszyńska et al. describe the immobilization of the Stenotrophomonas maltophilia KB2  strain, capable of degrading naproxen, on Luffa cylindrica used as a support.

The MS is well written, and several abiotic factors are considered for the imobilization of the strain.

Only few English language and style are fine/minor spell check are  required

Author Response

Reviewer comment:

The MS entitled "Immobilized Stenotrophomonas maltophilia KB2 in naproxen degradation" by Wojcieszyńska et al. describe the immobilization of the Stenotrophomonas maltophilia KB2  strain, capable of degrading naproxen, on Luffa cylindrica used as a support. The MS is well written, and several abiotic factors are considered for the imobilization of the strain. Only few English language and style are fine/minor spell check are  required.

Answer to Reviewer:

Thank you for your favorable review. The manuscript has been linguistically checked by a native speaker.

Reviewer 2 Report

This paper describes the optimum methods for immobilization of a bacteria and is used to degrade the naproxen in water. Some results are obtained, but they are not significant. In general, there is a lack of novelty in the study. The immobilization of bacteria is aimed at removing the pollutants from the water. However, the study on the removal of naproxen is less involved, and the removal efficiency was low. According to the previous study, the removal efficiecny of naproxen by Stenotrophomonas maltophilia KB2 can be reached to 78% in the presence of glucose, whereas the removal efficiency is about 25% in this study. Therefore, why the authors choose the immobilization technology to remove naproxen? In addition, the growth of microganisms or the metabolites after the removal of naproxen are not studied in this work.

Author Response

Reviewer Comments:

This paper describes the optimum methods for immobilization of a bacteria and is used to degrade the naproxen in water. Some results are obtained, but they are not significant. In general, there is a lack of novelty in the study. The immobilization of bacteria is aimed at removing the pollutants from the water. However, the study on the removal of naproxen is less involved, and the removal efficiency was low. According to the previous study, the removal efficiecny of naproxen by Stenotrophomonas maltophilia KB2 can be reached to 78% in the presence of glucose, whereas the removal efficiency is about 25% in this study. Therefore, why the authors choose the immobilization technology to remove naproxen? In addition, the growth of microganisms or the metabolites after the removal of naproxen are not studied in this work.

Answer for Reviewer:

Thank you for your comment and evaluation of our work. However, we cannot agree that this work adds nothing to the present knowledge. There are still few reports on the biodegradation of naproxen in the available literature, especially in the context of immobilized strains. On the other hand, naproxen is one of the most commonly used drugs that is manifested by its presence in the environment. Due to its multi-ring structure, it is also very stable. In turn, the KB2 strain we have given has unique properties not only for the degradation of naproxen but also for a wide range of highly toxic aromatic compounds usually found in wastewater treatment plants. Due to these unique properties, this strain has great application potential. However, due to the changing environmental conditions of wastewater treatment plants, the KB2 strain requires protection provided by the carrier. Although the comment of the reviewer regarding the lower efficiency of naproxen decomposition by the immobilized strain is correct, such an immobilized strain has a greater chance of functioning in the bioreactor environment than the free strain, and the concentrations appearing in sewage treatment plants are usually an order of magnitude smaller than those degraded by the immobilized KB2 strain.

Reviewer 3 Report

This manuscript discusses optimizing the immobilization parameters of microorganism on a luffa sponge as a carrier. The performance of immobilized KB2 strain was tested on Naproxen degradation and showed significant results with addition of carbon source. This manuscript is well written, and the findings were discusses clearly and in details.

I have few comments that could be considered to improve the final version of this manuscript:

- In Abstract, it is mentioned that "optical density of the culture 1.2", while in Figure 3 shows the highest biofilm content and enzymatic activity were obtained at OD 1.4. Please check

- Introduction, lines 21-28 add long sentences with no reference.

- line 59-61 "Immobilized microorganisms can be used in continuous processes or repeatedly in batch processes without the risk of washing them out of the bioreactor." for the current study, did the authors test the reuse-ability of the immobilized KB2 strains? how many cycles before start losing the enzymatic activity.

- line 99 "... compared to microorganisms introduced by bioaugmentation" add a reference.

- line 309, delete "2002)."

- Section 2.2, the results could be compared with free microorganism performance, to show that immobilization can improve the degradation of Naproxen.

- Section 3.2.2, add equation number for % biofilm formula.

Author Response

Reviewer Comment:

This manuscript discusses optimizing the immobilization parameters of microorganism on a luffa sponge as a carrier. The performance of immobilized KB2 strain was tested on Naproxen degradation and showed significant results with addition of carbon source. This manuscript is well written, and the findings were discusses clearly and in details.

Answer for Reviewer:

Thank you for your favorable review. 

Reviewer Comment:

I have few comments that could be considered to improve the final version of this manuscript:

In Abstract, it is mentioned that "optical density of the culture 1.2", while in Figure 3 shows the highest biofilm content and enzymatic activity were obtained at OD 1.4. Please check

Answer for Reviewer:

Thank you for your comment. In fact, an error has crept into the abstract. The optimal breeding density is 1.4. This was corrected in the manuscript.

Reviewer Comment:

Introduction, lines 21-28 add long sentences with no reference.

Answer for Reviewer:

Corrected as required.

Reviewer Comment:

Line 59-61 "Immobilized microorganisms can be used in continuous processes or repeatedly in batch processes without the risk of washing them out of the bioreactor." for the current study, did the authors test the reuse-ability of the immobilized KB2 strains? how many cycles before start losing the enzymatic activity.

Answer for Reviewer:

The authors did not test the immunized KB2 strain on a plant sponge for its multiple use. Such studies were conducted by the authors for another strain - Planococcus sp. S5 and were published in the work of Dzionek et al. (Catalysts 2018, 8, 176; doi:10.3390/catal8050176 ).

Reviewer Comment:

Line 99 "... compared to microorganisms introduced by bioaugmentation" add a reference.

Answer for Reviewer:

Corrected as required.

Reviewer Comment:

Line 309, delete "2002)."

Answer for Reviewer:

Corrected as required.

Reviewer Comment:

Section 2.2, the results could be compared with free microorganism performance, to show that immobilization can improve the degradation of Naproxen.

Answer for Reviewer:

The degradation of naproxen by free cells of the KB2 strain was published in Wojcieszyńska et al. (J. Environ. Manage. 145: 157-161, 2014), and the current studies indicate that the immobilization of the strain does not improve the degradation efficiency. However, it offers many other advantages, such as the protective effect of the carrier against adverse environmental factors.

Reviewer Comment:

Section 3.2.2, add equation number for % biofilm formula.

Answer for Reviewer:

Corrected as required.

Reviewer 4 Report

Pharmaceutical pollution is undoubtedly one of the most important environmental problems of our time. So, the manuscript on the degradation of naproxen, which is a popular non-steroidal anti-inflammatory drug often found in the environment, using immobilized bacterial biocatalysts, is of interest and value.

Comments:

-       lines 35-37: “Naproxen and its metabolites get into the treatment plant, however, the microbiome of wastewater treatment plants often does not completely degrade the drug. Hence they have been observed on effluents [4].” If the authors claim that naproxen is “often” not biodegradable in wastewater treatment plants, then there must be more than one reference.

-       Lines 60-63.  “Immobilized microorganisms can be used in continuous processes or repeatedly in batch processes without the risk of washing them out of the bioreactor. Immobilization ensures the stability of biochemical transformations, allows for their intensification by increasing the bioavailability of the substrate and specific enzymatic and respiratory activity, which reduces the costs of processes.” Please give some references here.

-       Lines 89-103: It seems that this part should be in Introduction.

-       Figure 2. “...depending on the time. ” Please clarify if it’s incubation time.

-       Figure 4. Can you guess why there is such a big difference in the biofilm content and the number of bacteria between pH 7.0 and pH 7.2?

-       Figures 1, 3, and 4. Please specify the day of the observations.

-       Lines 157-160. Too speculative. We can’t see the cell size from Figure 1A. Please rephrase.

-        The authors argue that “the starvation plays a major role in the adhesion of the strain on the support.” How to explain the increase in biomass seen in Figure 5. And visually, cells do not differ much in size depending on the incubation day .

-       Line 371 “only y 4.7 mg/L remained after 14 days”. Perhaps the authors meant 1.3 mg/L remained?

-       2.2. Naproxen degradation by immobilized KB2 strain. How well the authors enhanced the degradation of naproxen by immobilized KB2 cells compared to free KB2 cells (Wojcieszyńska et al.2014).

-       Line 388 “OD600 1.2”. But further, line 398 “OD600 of 0.2; 0.4; 0.6; 0.8; 1.0; 1.2 and 1.4,”. Please clarify.

-       Materials and methods. it is not clear how the authors calculated the CFU of immobilized cells. Did you deattach cells before calculating?

Author Response

Reviewer comment:

Pharmaceutical pollution is undoubtedly one of the most important environmental problems of our time. So, the manuscript on the degradation of naproxen, which is a popular non-steroidal anti-inflammatory drug often found in the environment, using immobilized bacterial biocatalysts, is of interest and value.

Answer for Reviewer:

Thank you for your favorable review. 

Reviewer comment:

Lines 35-37: “Naproxen and its metabolites get into the treatment plant, however, the microbiome of wastewater treatment plants often does not completely degrade the drug. Hence they have been observed on effluents [4].” If the authors claim that naproxen is “often” not biodegradable in wastewater treatment plants, then there must be more than one reference.

Answer for Reviewer:

Corrected as required.

Reviewer comment:

Lines 60-63.  “Immobilized microorganisms can be used in continuous processes or repeatedly in batch processes without the risk of washing them out of the bioreactor. Immobilization ensures the stability of biochemical transformations, allows for their intensification by increasing the bioavailability of the substrate and specific enzymatic and respiratory activity, which reduces the costs of processes.” Please give some references here.

Answer for Reviewer:

Corrected as required.

Reviewer comment:

Lines 89-103: It seems that this part should be in Introduction.

Answer for Reviewer:

The authors treat the indicated fragment as a preamble introducing the reader to the section discussing the results. It seems that such an introduction allows for a better understanding of the discussed results and is often practised in the scientific literature. For this reason, the authors decided to leave this excerpt in the Results and discussion section.

Reviewer comment:

Figure 2. “...depending on the time. ” Please clarify if it’s incubation time.

Answer for Reviewer:

Corrected as required.

Reviewer comment:

Figure 4. Can you guess why there is such a big difference in the biofilm content and the number of bacteria between pH 7.0 and pH 7.2?

Answer for Reviewer:

The obtained result is surprising also for the authors. Therefore, the experiment was repeated several times, obtaining similar results (the dispersion of the results is presented in the form of standard deviation in the chart). It is a phenomenon that is difficult to explain and cannot be justified by a change in surface charges. Moreover, the observed increase in biofilm at pH 7.2 is disproportionate to the increase in the number of cells. Hence, the observed phenomenon requires further multidirectional research.

Reviewer comment:

Figures 1, 3, and 4. Please specify the day of the observations.

Answer for Reviewer:

Corrected as required.

Reviewer comment:

Lines 157-160. Too speculative. We can’t see the cell size from Figure 1A. Please rephrase.

Answer for Reviewer:

The indicated fragment has been removed as you suggested.

Reviewer comment:

The authors argue that “the starvation plays a major role in the adhesion of the strain on the support.” How to explain the increase in biomass seen in Figure 5. And visually, cells do not differ much in size depending on the incubation day .

Answer for Reviewer:

Thank you for this comment. The ambiguity is related to the wrong place of introducing microscopic photos, which were taken from preparations obtained from immobilized bacteria used in the biodegradation of naproxen, where the carbon source was glucose and naproxen, enabling the growth of bacteria on the carrier. On the other hand, starvation conditions were used only during the immobilization process, when they stimulated biofilm formation (up to 72 hours). The fragment with the results of the SEM observation has been moved to the right place.

Reviewer comment:

Line 371 “only y 4.7 mg/L remained after 14 days”. Perhaps the authors meant 1.3 mg/L remained?

Answer for Reviewer:

Thank you for that rightful remark. The error has been corrected.

Reviewer comment:

2.2. Naproxen degradation by immobilized KB2 strain. How well the authors enhanced the degradation of naproxen by immobilized KB2 cells compared to free KB2 cells (Wojcieszyńska et al.2014).

Answer for Reviewer:

Immobilization did not improve the efficiency of naproxen degradation. However, it offers other additional benefits, such as protection of the strain against the toxic effects of xenobiotics or the possibility of multiple uses.

Reviewer comment:

Line 388 “OD600 1.2”. But further, line 398 “OD600 of 0.2; 0.4; 0.6; 0.8; 1.0; 1.2 and 1.4,”. Please clarify.

Answer for Reviewer:

We apologize for our oversight. The sentence "The bacterial suspension was introduced into the prepared medium so that the initial optical density of the culture (OD600) was 1.2." crept in by mistake. This sentence has been removed. Different OD values were used to determine the optimal culture optical density for the immobilization of the KB2 strain. 

Reviewer comment:

Materials and methods. it is not clear how the authors calculated the CFU of immobilized cells. Did you deattach cells before calculating?

Answer for Reviewer:

We apologize for this oversight. Yes, the cells were detached. A detailed description of CFU determination has been introduced in the Materials and methods chapter.

Round 2

Reviewer 2 Report

The responses can be partially accepted.However, the authors say the concentrations of naproxen in sewage treatment plants are lower than the concentration used in present study (6 mg/L). Therefore, the naproxen conentrations in sewage treatment plants and the application of immobilized technology in sewage treatment plants are better to add in the Introduction section. In addition, why the authors not select the naproxen at environment concentrations? 

Author Response

Reviewer comment

The responses can be partially accepted. However, the authors say the concentrations of naproxen in sewage treatment plants are lower than the concentration used in present study (6 mg/L). Therefore, the naproxen conentrations in sewage treatment plants and the application of immobilized technology in sewage treatment plants are better to add in the Introduction section. In addition, why the authors not select the naproxen at environment concentrations?

Answer for Reviewer

According to the recommendation, we have supplemented the Introduction with information related to naproxen concentrations in the inflows and outflows of sewage treatment plants. We also provided examples of the use of immobilized microorganisms in the degradation of naproxen under semi-technical conditions.

Reviewer comment

In addition, why the authors not select the naproxen at environment concentrations?

Answer for Reviewer

The concentration of 6 mg/l of naproxen was used in the research because the preparation is intended to strengthen the treatment process of naproxen-laden wastewater both in conditions of regular operation of the sewage treatment plant as well as in the event of large discharges loaded with naproxen. It is known that the loading of the sewage treatment plant is characterized by significant differences in naproxen concentrations, which may be even above 1 mg/l in a normally functioning sewage treatment plant. Research related to the degradation of naproxen by the KB2 strain in low concentrations, simulating the operating conditions of the strain treatment plant, is currently kept secret, related to the ongoing patent obtaining process.